# THSD7A Positivity Is Associated with High Expression of FAK in Prostate Cancer

**DOI:** 10.3390/diagnostics13020221

**Published:** 2023-01-07

**Authors:** Fidelis Andrea Flockerzi, Johannes Hohneck, Matthias Saar, Rainer Maria Bohle, Phillip Rolf Stahl

**Affiliations:** 1Department of Pathology, Saarland University Medical Center, 66421 Homburg, Germany; 2Department of Urology, University Hospital, 52074 Aachen, Germany

**Keywords:** THSD7A, FAK, prostate cancer

## Abstract

Prostate cancer is one of the most common malignancies, and there are a wide range of treatment options after diagnosis. Most prostate cancers behave in an indolent manner. However, a given sub-group has been shown to exhibit aggressive behavior; therefore, it is desirable to find novel prognostic and predictive (molecular) markers. THSD7A expression is significantly associated with unfavorable prognostic parameters in prostate cancer. FAK is overexpressed in several tumor types and is believed to play a role in tumor progression and metastasis. Furthermore, there is evidence that THSD7A might affect FAK-dependent signaling pathways. To examine whether THSD7A expression has an impact on the expression level of FAK in its unphosphorylated form, a total of 461 prostate cancers were analyzed by immunohistochemistry using tissue microarrays. THSD7A positivity and low FAK expression were associated with adverse pathological features. THSD7A positivity was significantly associated with high FAK expression. To our knowledge we are the first to show that THSD7A positivity is associated with high FAK expression in prostate cancer. This might be proof of the actual involvement of THSD7A in FAK-dependent signaling pathways. This is of special importance because THSD7A might also serve as a putative therapeutic target in cancer therapy.

## 1. Introduction

Prostate cancer is a common malignancy, having one of the highest incidences out of male cancers [1]. Though most prostate cancers behave in an indolent manner, a given sub-group exhibits an aggressive behavior, potentially leading to systemic disease and death [2,3]. For that reason, there are several treatment options for prostate cancer, ranging from watchful waiting and active surveillance to surgery, radiation, androgen deprivation therapy and chemotherapy [4]. Until now, prostate-specific antigen (PSA) levels in blood, the Gleason grade and tumor extension in biopsies have been the only established prognostic parameters in the preoperative setting. Given the wide range of therapeutic options and the concomitant side effects, novel predictive (molecular) markers are desirable.

In a former study, we analyzed the role of thrombospondin type-1 domain containing 7A (THSD7A) as a potential tumor antigen in cancer [5]. Among other findings, we demonstrated that THSD7A overexpression is significantly associated with unfavorable prognostic parameters in prostate cancer, including the tumor stage, Gleason grade and lymph node metastasis as well as PSA recurrence.

THSD7A is hypothesized to be a membrane-associated N-glycoprotein with a soluble form produced by cells of endothelial and neuronal origin. The soluble form can promote endothelial cell migration, tube formation and sprouting in angiogenesis [6,7,8]. Furthermore, THSD7A is involved in the pathogenesis of membranous nephropathy [9,10,11,12]. In addition to our previous results, several other reports indicate that THSD7A might play a role in different tumor types [13,14,15,16,17,18]. Furthermore, several other groups investigated the possible relationship between THSD7A expression in malignant diseases and the development of membranous nephropathy [13,14,19,20,21,22,23,24,25].

Focal adhesion kinase (FAK) in turn is quite an established tumor marker. FAK is a protein tyrosine kinase that regulates cellular adhesion, motility, proliferation and survival in various types of cells. FAK is overexpressed in several tumor types and is believed to play a role in tumor progression and metastasis [26,27,28,29]. Therefore, FAK is discussed as an effective cancer target in various tumors [30,31,32,33,34].

Several studies deal with the role of FAK in prostate cancer but only a few groups report the expression status of FAK in this cancer, and the results seem somewhat ambiguous [35,36,37,38]. The three major non-receptor tyrosine kinases, FAK, SRC and ETK, form the SRC tyrosine kinase complex, which in prostate cancer is suggested to play an important role in the aberrant activation of the androgen receptor (AR) by phosphorylation [39]. There is also evidence that FAK activation might be an important factor in androgen-independent progression to neuroendocrine carcinoma [40].

There is evidence that THSD7A might affect FAK-dependent signaling pathways [7,8]. The main objective of this study was to examine whether the THSD7A expression status has an impact on the expression level of FAK in its unphosphorylated form. The secondary objective was to examine the association of FAK in its unphosphorylated form and the common pathological parameters in prostate cancer.

Therefore, a total of 461 prostate cancers were analyzed by immunohistochemistry (IHC) using tissue microarrays (TMAs).

## 2. Materials and Methods

### 2.1. Tissue Samples

Tissue samples were derived from primary surgically resected prostatectomy specimens (*n* = 461). All patients were treated at the Department of Urology at the Saarland University Medical Center, Saarland University, Homburg/Saar, Germany between 2012 and 2020. For all tumors, detailed histopathological data on Gleason grade and pT-status s were available; six patients lacked nodal status (Table 1 and Table 2). Exclusion criteria for the cohort was having had neoadjuvant therapy. Tissue samples of the resected prostatectomy specimen were analyzed by immunohistochemistry (IHC) using tissue microarrays (TMAs).

### 2.2. Tissue Microarrays

TMA construction was performed using a manual tissue arrayer and according to the manufacturer’s directions (Manual Tissue Arrayer, AlphaMetrix Biotech, Roedermark, Germany). Tissue cylinders each with a diameter of 0.6 mm were punched out of selected paraffin-embedded tumor tissue blocks and were brought into empty “recipient” paraffin blocks. Four µm sections of the TMA blocks were transferred to adhesion slides (Matsunami TOMO) and were used for immunohistochemistry (IHC).

### 2.3. Immunohistochemistry

The TMA blocks were cut into 4 µm sections, transferred to adhesive slides (Matsunami TOMO) using a water bath (46 °C) and dried overnight at 37 °C. Staining was performed with Benchmark Ultra (Ventana Medical Systems) using primary antibody specific for THSD7A (rabbit polyclonal antibody, Sigma Aldrich, St. Louis, MO, USA; cat# HPA000923; dilution 1:150) and FAK (monoclonal antibody (clone 4.47), Millipore; cat# 05-537; dilution 1:100).

Bound antibody was then visualized using the ultraView Universal Alkaline Phosphatase Red Detection (Roche, Basel, Switzerland) according to the manufacturer’s directions. Heat-induced antigen retrieval at 97 °C was performed with CC2 buffer (Ventana) for 56 min (THSD7A) and with CC1 buffer (Ventana) for 64 min, respectively.

For evaluation of THSD7A expression, the percentage of positive cells was estimated, and the staining intensity was recorded semiquantitatively as 0, 1+, 2+ or 3+ for each tissue sample. The staining results were categorized into the following four groups for statistical analysis: tumors without any staining were considered negative; tumors with 1+ staining in ≤70% or with 2+ staining in ≤30% of cells were considered weakly positive; tumors with 1+ staining in >70% and with 2+ staining in >30% but ≤70% and with 3+ staining in <30% of cells were considered moderately positive; and tumors with 2+ staining in >70% and with 3+ staining in ≥30% of cells were considered strongly positive. To better define THSD7A expression, we dichotomized THSD7A expression as negative (no staining in any tumor cell) and positive (at least 1+ staining in at least a few tumor cells).

For evaluation of FAK expression, the percentage of positive cells was estimated, and the staining intensity was recorded semiquantitatively as 0, 1+, 2+ or 3+ for each tissue sample. To better define FAK expression, we dichotomized FAK expression as low (tumors with 0 staining, 1+ staining, 2+ staining in ≤70% and 3+ staining in ≤30% of cells) or high (tumors with 2+ staining in >70% of cells and 3+ staining in >30% off cells).

### 2.4. Statistics

Statistical analysis was performed using R (R Corporation 2021, R Foundation for Statistical Computing). Fisher’s exact test was used for testing the null hypothesis of independence. For cross tables larger than 2 × 2, pairwise Fisher’s tests and *p*-value adjustment via Benjamini–Hochberg procedure were performed as post-hoc analysis.

Regarding Gleason score, tumors were split up into the five distinct grade groups according to the World Health Organization (grade group 1: *n* = 96; grade group 2: *n* = 98; grade group 3: *n* = 124; grade group 4: *n* = 123; grade group 5: *n* = 20).

## 3. Results

A total of 397 (86.1%) tumors were analyzable for THSD7A-IHC. Sixty-four cases were not analyzable due to a lack of tissue in the TMA spot or due to a lack of unequivocal tumor tissue. A total of 41 (10.3%) tumors showed at least weak positivity with mainly cytoplasmic staining but also membranous staining in some cases. Less than 5% of non-tumor prostate tissue revealed very weak cytoplasmic THSD7A expression. Representative images are shown in Figure 1. THSD7A positivity was associated with advanced tumor stage (*p* < 0.001), positive nodal status (*p* < 0.001) and with a high Gleason score (*p* < 0.001). The results are shown in detail in Table 1.

A total of 361 (78.3%) tumors were analyzable for FAK-IHC. One hundred cases were not analyzable due to a lack of tissue in the TMA spot or due to a lack of unequivocal tumor tissue. A total of 222 (61.4%) tumors showed low expression and a total of 139 (38.6%) tumors showed high expression with a cytoplasmic staining pattern, respectively. Non-tumor prostate tissue revealed low cytoplasmic FAK expression. Representative images are shown in Figure 2. Low FAK expression was associated with advanced tumor stage (*p* = 0.007) and positive nodal status (*p* = 0.005). While there seemed to be a significant association of FAK expression with the Gleason score (*p* = 0.002), the post-hoc analysis revealed a significant difference only between groups 2 and 3. The results are shown in detail in Table 2.

A total of 345 (74.8%) tumors were analyzable for FAK-IHC as well as for THSD7A-IHC. THSD7A positivity was significantly associated with high FAK expression (*p* = 0.003). The results are shown in detail in Table 3. Figure 3 shows a tumor with high FAK expression and negativity for THSD7A. Figure 4 shows a tumor with high FAK expression and positivity for THSD7A.

## 4. Discussion

Prostate cancer is one of the most common malignancies. However, the vast majority of patients with diagnoses of prostate cancer will not die from the disease. Due to the fact that there are a wide range of treatment options for prostate cancer and that some of them have concomitant side effects, it is desirable to find novel prognostic and predictive (molecular) markers.

Several studies indicate that THSD7A might play a role at least in the prognosis of different tumor types [5,13,15,16,17,23]. FAK is believed to play an important role in prostate cancer and is discussed as a potential therapeutic target, especially in advanced stages [34,41,42,43,44,45]. Moreover, there is evidence that FAK-dependent signaling pathways might be affected by THSD7A [7,8]. For this reason, we wanted to examine whether THSD7A expression status has an impact on the expression level of FAK in its unphosphorylated form.

In our analysis, as one could expect from previous investigations, THSD7A positivity was associated with adverse pathological features. Surprisingly, low FAK expression was associated with an advanced tumor stage and nodal metastasis. We did not expect this correlation, given the large number of studies which describe FAK overexpression in several tumor types and its potential role in tumor progression and metastasis. However, as FAK is a protein tyrosine kinase, it exists in a unphosphorylated/inactivated form and a phosphorylated/activated form. Most of the studies reporting on the above-mentioned correlation dealt with the phosphorylated/activated form of FAK. Though FAK is an established tumor marker and a potential therapeutic target, few data can be found on FAK’s status in prostate cancer regarding its expression quantity, and the reported results are ambiguous.

Rovin et al. [37] did not find an association of FAK expression or staining intensity with the tumor grade or tumor stage using immunohistochemistry when investigating human prostate specimens. Tremblay et al. [35] in contrast stated that an increase in FAK mRNA and protein correlates with progression and invasion in prostate cancer. However, this group investigated a rather small number (*n* < 100) of samples. Furthermore, the samples included cell lines and human tissue obtained from patients undergoing transurethral resection or from autopsies. Slack JK et al. [36] also report that an increased metastatic potential in prostate cancer correlates with increased FAK expression, though this assumption was solely a result of cell line analysis.

Several different clones for both forms of FAK are available (we used clone 4.47 which detects the unphosphorylated/inactivated form of FAK). Since there exist great differences in utilized methods, materials, protocols, clones, etc., in studies dealing with FAK expression, deviations in the results can be expected. So the final results do not necessarily contradict each other.

Since there is evidence that FAK-dependent signaling pathways might be affected by THSD7A, we first focused on FAK in its unphosphorylated form.

It is all the more impressive that THSD7A positivity was significantly associated with high FAK expression, given the inverse correlation of THSD7A and FAK expression levels with common pathological features.

To our knowledge, we are the first to show that THSD7A positivity is associated with high FAK expression in prostate cancer. There is evidence that FAK might be activated by THSD7A. Our findings show that FAK expression levels might be raised by THSD7A and may be proof of the actual involvement of THSD7A in FAK-dependent signaling pathways. This is of special importance because THSD7A, as a membrane-associated protein, might serve as a putative therapeutic target in cancer therapy.

The limitations of this study were that we only used immunohistochemistry to determine expression levels. For that reason, we do not have reliable information on what caused the determined expression levels of the investigated markers (for example, genetic alterations). We examined the expression level of FAK in its unphosphorylated form. Finally, it is not possible to make a valid statement whether THSD7A effectively raises FAK expression levels or plays a role in the activation/phosphorylation of FAK. Further studies are necessary to clarify the described connection.

## 5. Conclusions

To our knowledge, we are the first to show that THSD7A positivity is associated with high FAK expression in prostate cancer. THSD7A is a membrane-associated protein. Given the potential involvement of THSD7A in FAK-dependent signaling pathways, THSD7A should be discussed as a therapeutic target in prostate cancer.

## Figures and Tables

**Figure 1 diagnostics-13-00221-f001:**
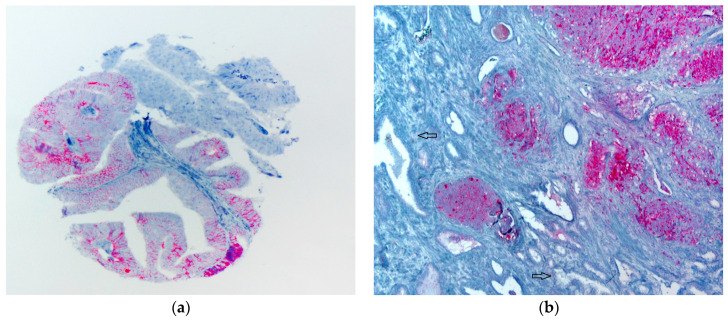
THSD7A shows membranous and cytoplasmic staining pattern. (**a**) Tumor cells with 2+ and 3+ staining. (**b**) Whole slide, tumor cells with strong THSD7A positivity, and the adjacent non-tumor tissue is THSD7A-negative (arrows) with a magnification of 100×.

**Figure 2 diagnostics-13-00221-f002:**
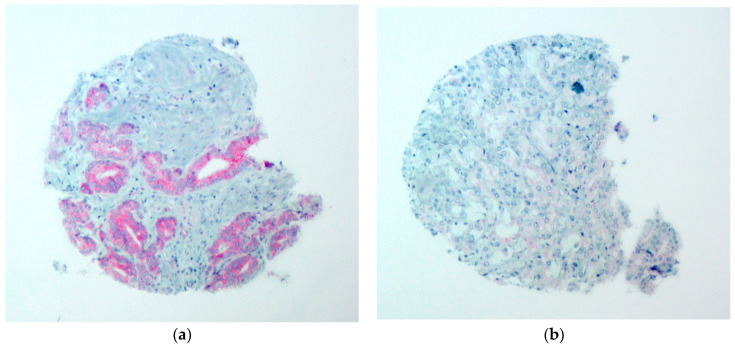
FAK shows cytoplasmic staining pattern; (**a**) tumor cells with high FAK expression; (**b**) tumor cells with low FAK expression with a magnification of 100×.

**Figure 3 diagnostics-13-00221-f003:**
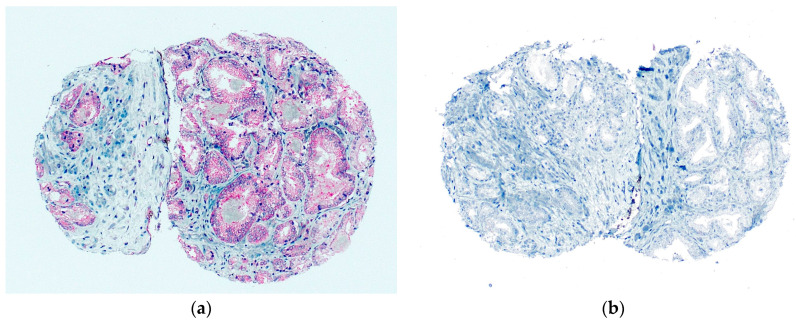
(**a**) Tumor with high FAK expression; (**b**) same tumor is THSD7A-negative with a magnification of 100×.

**Figure 4 diagnostics-13-00221-f004:**
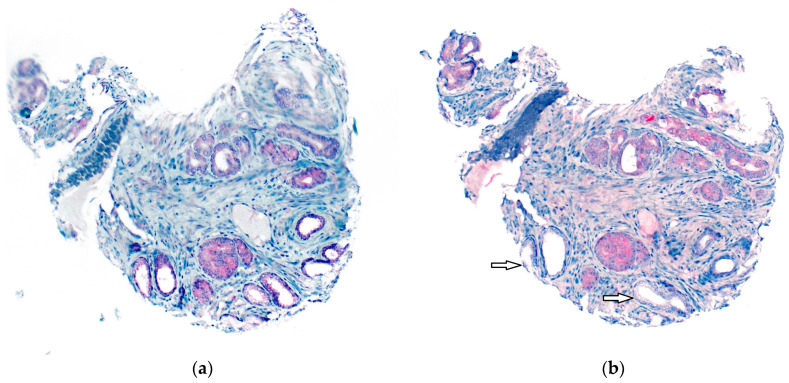
(**a**) Tumor with high FAK expression; (**b**) same tumor is also THSD7A-positive and the adjacent non-tumor tissue is THSD7A-negative (arrows) with a magnification of 100×.

**Table 1 diagnostics-13-00221-t001:** Association between THSD7A expression and common pathological parameters.

	THSD7A		*p*	OR (95% KI)
	Negative	Positive		
pT-Status				
pT2	180	9	<0.001	3.62 (1.63; 8.89)
pT3–pT4	176	32		
Nodal Status				
N0	273	14	<0.001	6.40 (3.08; 14.08)
N+	78	26		
WHO Grade Group				
1	73	1	<0.001	
2	80	6		
3	84	15		
4	106	12		
5	13	7		

**Table 2 diagnostics-13-00221-t002:** Association between FAK-expression and common pathological parameters.

	FAK		*p*	OR (95% KI)
	Low	High		
pT-Status				
pT2	88	76	0.007	0.55 (0.35; 0.86)
pT3–pT4	134	63		
Nodal Status				
N0	145	110	0.005	0.48 (0.27; 0.81)
N+	74	26		
WHO Grade Group				
1	24	30	0.002	
2	38	42		
3	71	26		
4	75	35		
5	14	6		

**Table 3 diagnostics-13-00221-t003:** Association between FAK expression and THSD7A expression.

	FAK		*p*	OR (95% KI)
	Low	High		
THSD7A				
negative	198	108	0.003	2.93 (1.49; 6.27)
positive	15	24		

## Data Availability

The datasets used and analyzed in this paper are available from the corresponding author on reasonable request.

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
