# Peer review of "THSD7A Positivity Is Associated with High Expression of FAK in Prostate Cancer"

_diagnostics, 2023, doi:10.3390/diagnostics13020221_

Round 1

Reviewer 1 Report

Dear Academic Editor:

I have reviewed the study carried out by Flockerzi F.D., et al., entitled: “THSD7A-positivity is associated with high expression of FAK 2 in prostate cancer”. In brief, The study focuses on analyzing the immunohistochemical expression of THSD7A and FAK in 461 prostate cancers. This is a topic that could be interesting to the journal readership. Here are several comments and suggestions for improvement:

Introduction:

1)    The introduction should highlight the main objective of the study and the secondary objectives.

2)    References must be corrected according to the established format.

Materials & Methods:

1)    Two descriptive tables are used, without heading they are shown in this section. They must be located in results, with an adequate description.

2)    Line: 73-74: For all tumors detailed histopathological data on Gleason grade, pT-status and pN-73 status were available (Table 1 and Table 2). What is the pathological stage (pT) pT2c?. The sample studied is based on samples from prostatectomies, and how is pT1 explained?

3)    Authors should specify in the (WHO) 2 and 3 grade groups, the percentage of Gleason 4 pattern and whether the cribriform pattern is present or absent.

4)    What were the inclusion and exclusion criteria.

5)    The study has not been approved by an ethics committee?.

Results:

1)    Line 120: A total of 397 (86.1%) tumors were analyzable…. Why the difference between the N of materials and methods, and that expressed in results? For what is this?

2)    Line 127: The results are shown in detail in Table 1. The table should be placed after the statement, with its description.

3)    Line 125-126: Regarding Gleason-score, post-hoc analysis yielded significant results for pairwise comparison between group 1 and 3 as well as group 5 and the groups 1, 2 and 4. I don't understand.

4)    Line 132: A total of 361 (78.3%) tumors were analyzable for FAK-IHC. Why the difference between the N of materials and methods, and that expressed in results? For what is this?.

5)    The quality of the images should be improved.

Discussion:

1)    Line 161-166: In our analysis, as one could expect from previous investigations, THSD7A-positivity was associated with adverse pathological features. Surprisingly, low FAK expression was associated with advanced tumor stage and nodal metastasis. We did not expect this correlation, given the large number of studies which describe FAK overexpression in several tumor types and its potential role in tumor progression and metastasis. However, several different clones were used to detect FAK in these studies, mainly clones to detect the activated/phosphorylated form of FAK. This point needs further discussion. If different clones show different and variable expressions, can they indicate different levels of expression? Why did the authors not evaluate other clones in the study?.

2)    Line 180-181: Besides, different clones were used in these studies to detect FAK (we used clone 4.47 which detects the unphosphorylated/inactivated form of FAK), so finally the results do not necessarily contradict each other.

3)    It is not disputed whether there were differences between the confined organ and the non-confined organ.

4)    What were the limitations of the study?.

5)    The discussion should be rewritten and expanded.

Author Response

We thank Reviewer 1 for his comments and tried to answer the requests in an appropriate way.

Introduction:

1)    The introduction should highlight the main objective of the study and the secondary objectives.

Reviewer 1 asked us to highlight the main objective of the study and the secondary objectives. The main objective of course was to examine whether THSD7A-expression status has an impact on the expression level of FAK in its unphosphorylated form. Secondary objective was to examine the association between FAK in its unphosphorylated form and common pathological parameters in prostate cancer. This was highlighted at the end of the Introduction (Line: 62-65).

2)    References must be corrected according to the established format.

As requested by reviewer 1, the references were corrected according to the established format.

Materials & Methods:

1)    Two descriptive tables are used, without heading they are shown in this section. They must be located in results, with an adequate description.

As requested by reviewer 1, Table 1 and Table 2 received an adequate description.

We absolutely agree with reviewer 1 that the tables are better placed in the Results section. In the submitted manuscript they all were put in the Results section. I guess it was a decision by the editing staff to put Table 1 and Table 2 in the Materials and Methods section.

2)    Line: 73-74: For all tumors detailed histopathological data on Gleason grade, pT-status and pN-73 status were available (Table 1 and Table 2). What is the pathological stage (pT) pT2c?. The sample studied is based on samples from prostatectomies, and how is pT1 explained?

Reviewer 1 pointed out to us that the description of the tumor stage in Table 1 and Table 2 is equivocal. He also asked for the meaning of pT2c as well as the explanation of pT1-status in our cohort of prostatectomies. Tumor stage corresponds to the pT-status. pT2c-status defines a tumor which is found in both prostate lobes but is limited to the prostate. pT1-status does not exist for prostate cancer. We made corrections in Table 1 and Table 2.

3)    Authors should specify in the (WHO) 2 and 3 grade groups, the percentage of Gleason 4 pattern and whether the cribriform pattern is present or absent.

Reviewer 1 asked us to specify the percentage of Gleason 4 pattern in the WHO grade groups 2 and 3 and to indicate whether a cribriform pattern is present or absent.

As this is a retrospective study we had to deal with the information we got through our pathology reports. In many of these partly older reports we can not find information on the percentage of Gleason 4 pattern, Furthermore, information on the presence of cribriform Gleason pattern can at best be found in recently reports. To provide reliable information on this issue it would be necessary to review almost 200 cases (approximately 8000-10000 slides). This would be possible if it is considered absolutely necessary.

We are surely aware that the percentage of Gleason 4 pattern as well as the presence of cribriform pattern have an impact on prognosis in prostate cancer. However, there already exists a 5-tired grading system by the WHO. Specifying the percentage of Gleason 4 pattern and indicating whether a cribriform pattern is present would create several more subgroups. For statistical reasons there might occur severe feasibility problems

4)    What were the inclusion and exclusion criteria.

Reviewer 1 asked for inclusion and exclusion criteria. The only exclusion criteria was neoadjuvant therapy. We added this information in the Materials and Methods section (Line: 74-75).

5)    The study has not been approved by an ethics committee?.

The study has been approved by the ethics committee of the Medical chamber of the Saarland, Germany (Identification number: 282/19) This information was added (Line 248-249).

Results:

1)    Line 120: A total of 397 (86.1%) tumors were analyzable…. Why the difference between the N of materials and methods, and that expressed in results? For what is this?

Reviewer 1 asked why there is a difference between total case number and analyzable cases. A total of 64 cases were not analyzable due to a lack of tissue in the TMA spot or due to a lack of unequivocal tumor tissue. This information was added to the Results section (Line: 122-124)

2)    Line 127: The results are shown in detail in Table 1. The table should be placed after the statement, with its description.

Table 1 received an adequate description.

We absolutely agree with reviewer 1 that Table 1 is better placed in the Results section. In the submitted manuscript we put Table 1 in the Results section. I guess it was a decision by the editing staff to put Table 1 in the Materials and Methods section.

3)    Line 125-126: Regarding Gleason-score, post-hoc analysis yielded significant results for pairwise comparison between group 1 and 3 as well as group 5 and the groups 1, 2 and 4. I don't understand.

Reviewer 1 asks us for an explanation of the equivocal annotations concerning the association between Gleason-score and THSD7A-expression.

Exact Fisher’s test yielded a significant difference in THSD7a expression over all 5 subgroups. To check if which of the 5 subgroups differ in THSD7a expression post analysis was performed using pairwise Fisher’s test und alpha correction (via Benjamini-Hochberg procedure).

To simplify the correlation we deleted the above mentioned sentence (Line: 129-130).

4)    Line 132: A total of 361 (78.3%) tumors were analyzable for FAK-IHC. Why the difference between the N of materials and methods, and that expressed in results? For what is this?.

Reviewer 1 asked why there is a difference between total case number and analyzable cases. A total of 100 cases were not analyzable due to a lack of tissue in the TMA spot or due to a lack of unequivocal tumor tissue. This information was added to the Results section (Line: 139-141)

5)    The quality of the images should be improved.

Reviewer 1 asked us to improve the quality of the images. We provided more figures in the required resolution and labeled different areas (Figure1-4).

Discussion:

1)    Line 161-166: In our analysis, as one could expect from previous investigations, THSD7A-positivity was associated with adverse pathological features. Surprisingly, low FAK expression was associated with advanced tumor stage and nodal metastasis. We did not expect this correlation, given the large number of studies which describe FAK overexpression in several tumor types and its potential role in tumor progression and metastasis. However, several different clones were used to detect FAK in these studies, mainly clones to detect the activated/phosphorylated form of FAK. This point needs further discussion. If different clones show different and variable expressions, can they indicate different levels of expression? Why did the authors not evaluate other clones in the study?.

2)    Line 180-181: Besides, different clones were used in these studies to detect FAK (we used clone 4.47 which detects the unphosphorylated/inactivated form of FAK), so finally the results do not necessarily contradict each other.

1) and 2)

Reviewer 1 asked for further discussion concerning varying results in different studies dealing with FAK expression. For that reason, we deepened the discussion on that issue (Line: 187-192, 205-212)

3)    It is not disputed whether there were differences between the confined organ and the non-confined organ.

Reviewer 1 asked us to provide information on differences between the confined organ and the non-confined organ. There was a significant association between both markers respectively with organ limited (pT2-status) and not organ confined (pT3 -and pT4-status) disease. These statements can be found in the Results section and in Table 1 and Table 2.

4)    What were the limitations of the study?

Reviewer 1 asked for the limitations of the study. We provide this information in Line: 222-228

5)    The discussion should be rewritten and expanded.

Reviewer 1 asked us to rewrite and expand the discussion.

The discussion was rewritten and expanded according to the points of criticism of reviewer 1.

Reviewer 2 Report

The authors investigated the connection between THSD7A and FAK expression in prostate cancer. Nonetheless, there are concerns that must be addressed. 

1. The authors should provide detailed cohort information. How many samples lacked Nodal classification and WHO grade? Since the total number of patients in Tables 1 and 2 differs. Is there a relationship between THSD7A and PSA? 

2. The author's presentation of IHC staining for non-tumor and tumor tissue is commendable. It may be beneficial to label these regions in Figure 1. In addition, the low or absence of THSD7A staining in tumor cells should be presented. 

3. The author confirms the significance of the relationship between FAK and THSD7A expression in prostate cancer using the Fisher test. To demonstrate the significant relationship between FAK and THSD7A expression, a representative IHC staining of FAK and THSD7A in the same patient should be presented. Are FAK and THSD7A staining scores correlated (Pearson correlation test)? Is there a significant correlation between mRNA levels of FAK and THSD7A in TCGA prostate cohort?

Author Response

We thank Reviewer 2 for his comments and tried to answer the requests in an appropriate way.

  1. The authors should provide detailed cohort information. How many samples lacked Nodal classification and WHO grade? Since the total number of patients in Tables 1 and 2 differs. Is there a relationship between THSD7A and PSA? 

Reviewer 2 asked for detailed cohort information since he found some irregularities in Table 1 and Table 2. A total of six patients did not receive lymphonodecty and therefore lacked nodal status (Nx). The only exclusion criteria for the cohort was neoadjuvant therapy. For this reason, for all patients Gleason-score/WHO grade was available. We checked our raw data. The irregularities resulted from transmission errors to the tables. They did not have an impact on statistical analysis.

As this is a retrospective study we had to deal with the information we got through our pathology reports. For most patients PSA-levels were not available from the pathology reports. For that reason, we can not provide information on the relationship between THSD7A and PSA.

Additional information was provided in the Materials and Methods section (Line 73-75) and corrections were made in Table 1 and Table 2.

  1. The author's presentation of IHC staining for non-tumor and tumor tissue is commendable. It may be beneficial to label these regions in Figure 1. In addition, the low or absence of THSD7A staining in tumor cells should be presented.

Reviewer 2 asked us to label tumor tissue and adjacent non-tumor tissue in Figure1. Furthermore, the reviewer asked us to present tumor tissue with low staining intensity or absent THSD7A-staining in tumor cells. These changes were made and are presented in Figure1b, Figure3b, Figure4b.

  1. The author confirms the significance of the relationship between FAK and THSD7A expression in prostate cancer using the Fisher test. To demonstrate the significant relationship between FAK and THSD7A expression, a representative IHC staining of FAK and THSD7A in the same patient should be presented. Are FAK and THSD7A staining scores correlated (Pearson correlation test)? Is there a significant correlation between mRNA levels of FAK and THSD7A in TCGA prostate cohort?

Reviewer 2 asks us to present a representative IHC staining of FAK and THSD7A in the same patient. These images are presented in Figure4.

Reviewer 2 asks for a Pearson correlation test regarding the results of FAK and THSD7A. We are not quite sure whether the requirements for Pearson correlation test are met for this issue. Perhaps more information concerning this problem could help us to understand the intension.

Round 2

Reviewer 1 Report

The authors have responded to all my comments. I acknowledge that the authors have rewritten some parts of the manuscript, taking into consideration the suggestions made.

Reviewer 2 Report

Accpet in present form.